# Physiological Features of Olympic-Distance Amateur Triathletes, as Well as Their Associations with Performance in Women and Men: A Cross–Sectional Study

**DOI:** 10.3390/healthcare11040622

**Published:** 2023-02-20

**Authors:** José Geraldo Barbosa, Claudio Andre Barbosa de Lira, Rodrigo Luiz Vancini, Vinicius Ribeiro dos Anjos, Lavínia Vivan, Aldo Seffrin, Pedro Forte, Katja Weiss, Beat Knechtle, Marilia Santos Andrade

**Affiliations:** 1Department of Physiology, Federal University of São Paulo, São Paulo 04021-001, Brazil; 2Human and Exercise Physiology Division, Faculty of Physical Education and Dance, Federal University of Goiás, Goiânia 74690-900, Brazil; 3Center for Physical Education and Sports, Federal University of Espírito Santo, Vitória 29075-210, Brazil; 4Department of Sports, Higher Institute of Educational Sciences of the Douro, 4560-547 Porto, Portugal; 5Department of Sports Sciences, Instituto Politécnico de Bragança, 5300-253 Bragança, Portugal; 6Research Center in Sports, Health and Human Development, 7000-671 Covilhã, Portugal; 7Institute of Primary Care, University of Zurich, 8091 Zurich, Switzerland; 8Medbase St. Gallen Am Vadianplatz, 9000 St. Gallen, Switzerland

**Keywords:** triathlon, ports physiology, performance, women, maximal aerobic speed

## Abstract

The purpose of this study was to verify the physiological and anthropometric determinants of triathlon performance in female and male athletes. This study included 40 triathletes (20 male and 20 female). Dual-energy X-ray absorptiometry (DEXA) was used to assess body composition, and an incremental cardiopulmonary test was used to assess physiological variables. A questionnaire about physical training habits was also completed by the athletes. Athletes competed in the Olympic-distance triathlon race. For the female group, the total race time can be predicted by V̇O_2_max (β = −131, t = −6.61, *p* < 0.001), lean mass (β = −61.4, t = −2.66, *p* = 0.018), and triathlon experience (β = −886.1, t = −3.01, *p* = 0.009) (r^2^ = 0.825, *p* < 0.05). For the male group, the total race time can be predicted by maximal aerobic speed (β = −294.1, t = −2.89, *p* = 0.010) and percentage of body fat (β = 53.6, t = 2.20, *p* = 0.042) (r^2^ = 0.578, *p* < 0.05). The variables that can predict the performance of men are not the same as those that can predict the triathlon performance of women. These data can help athletes and coaches develop performance-enhancing strategies.

## 1. Introduction

Although female participation in triathlon is still lower than male participation (25–40%), there has been a significant increase in female participation in this sport since 1990 [1,2,3]. Female participation has increased not only in triathlon, but also in several other sports. In running, female participation reached the same percentage as male participation in 2018, and at the most recent Olympic Games (Tokyo 2020), female participation set a new record, reaching 49% [4].

Despite a recent surge of female participation in sports, most scientific studies on sports sciences continue to focus on men [4]. Therefore, the results of the studies conducted on male athletes on sports training are applied to both male and female athletes, despite the lack of a reasonable scientific justification [5,6].

The literature’s consensus that maximal oxygen uptake (V̇O_2_max), the percentage of V̇O_2_max that can be sustained for an extended period of time, the running economy, and body composition are important variables associated with performance in long-distance events [7]. However, the relative importance of each one can differ for male or female performance, once there are several physiological (red cell mass, hemoglobin, muscular fiber type percentage, muscle capillarization, vasodilatory capacity, and energetic substrate use) and body composition (fat mass percentage, lean mass) differences between sexes [8]. The variables associated with triathlon performance have previously been studied. Therefore, prediction equations for triathlon performance have also been developed; however, previous studies have only included male athletes, or when female athletes were included, the sex was not evaluated as a biological variable [9,10]. The knowledge of the main predictive variables of performance for each sex separately, can help coaches in directing the training sessions to obtain adaptive responses of the most important predictive variables of performance and optimize the improvement of sports performance.

Therefore, the purpose of this study was to confirm the levels of association between physiological and body composition variables and performance in an Olympic-distance triathlon test for each sex and, later, to describe a performance prediction equation for the modality based on the measured variables for each sex. Second, the study aimed to compare physiological and body composition variables between sexes. We hypothesized that the level of association between the measured variables and triathlon race results would differ between sexes, and thus the Olympic-distance triathlon race prediction equations would differ for each sex.

## 2. Materials and Methods

### 2.1. Participants

Triathletes were invited to take part in the study via social media (WhatsApp, Instagram, and email), as well as folders distributed at triathlon competitions. The inclusion criteria for participating in the study were being enrolled in the 30th Santos International Olympic Triathlon in February 2022, between the ages of ≥18 and no more than 61 years, both sexes, training triathlon regularly for at least 6 months, and having a medical allowance. The exclusion criteria included being pregnant, having competed in the alternate modality, failure to finish the race, or failure to submit to laboratory tests for any reason. Initially, 42 athletes were selected to participate in the study (22 men and 20 women). Two male athletes were excluded from the study, one was due to the fact that he did not finish the race and one was due to the fact that he failed to submit to laboratory tests. As a result, the study includes 20 male athletes and 20 female athletes. Data were collected during the pre-season. Table 1 shows the descriptive characteristics of the sample.

### 2.2. Procedures

All experimental procedures followed the Declaration of Helsinki and the Recommendations for the Conduct, Reporting, Editing, and Publication of Scholarly Work in Medical Journals. The study was approved by the Human Ethics Committee of the University Federal of São Paulo-UNIFESP (approval number 5.059.538, 25 October 2021). The participants were given information about the purpose of the research, all of the proposed physiological laboratory tests, and the risks and benefits. The researchers justified the principles of respect for the volunteers, as well as the guarantee of privacy, confidentiality, and anonymity rights. After all, every participant signed the informed consent form. Initially, the volunteers completed an online questionnaire from the Google Forms Platform about their training habits and medical conditions. Then, they attend the Exercise Physiology Laboratory at UNIFESP once, during the morning period. During the visit, volunteers were measured for height, body mass, and body composition. Thereafter, participants were submitted to a running economy evaluation. After 30 min of rest, participants were subjected to a cardiorespiratory maximal treadmill test. They were instructed to abstain from strenuous training in the last 24 h before the test and not consume hyper-stimulating foods on the day (e.g., caffeine). Wearing light clothes and comfortable running shoes was also recommended.

The organizers provided the results of the total race time and split times of the competition, which were taken from the official website of the event (https://www.internacionaldesantos.com.br, accessed on 5 March 2022). All the data were collected in January 2022 (pre-competition phase), 1 month before the 30th Santos International Olympic Triathlon. 

### 2.3. Questionnaire

The questionnaire includes two open questions about their medical condition: Do you have any chronic diseases? Do you take any medications? The questionnaire also includes four open questions about their training habits: How many hours per week do you cycle train? How many hours per week do you train for running? How many hours per week do you train for swimming? How many days, months, or years have you been training for a triathlon?

### 2.4. Morphological Variables

Body mass and height were measured to the nearest 0.1 kg and 0.1 cm using a calibrated stadiometer Filizola^®^ PL (Filizola, São Paulo, SP, Brazil), respectively. Body composition was determined using dual-energy X-ray absorptiometry (DEXA, software version 12.3, Lunar DPX, GE Healthcare, Madison, WI, USA). The volunteers were instructed to drink water ad libitum and were not given any instructions about fasting or taking any specific feedings prior to the procedure. They were all evaluated after bladder voiding [11]. These procedures had been shown to be a reliable method for assessing body composition [12,13].

### 2.5. Running Economy Test

The volunteers were subjected to a treadmill running test for 4 min on a motorized treadmill (Inbrasport, ATL, Porto Alegre, Brazil) using a computer-based breath, via a breath gas exchange analyzer (Quark, Comedy, Italy) at a constant speed of 8 km/h, which was below the ventilatory threshold (VT) for all volunteers. Prior to each test, the calibration procedure was carried out according to the manufacturer’s instructions. The last minute was taken into account when calculating the average oxygen uptake, CO_2_ production, and respiratory exchange rate (RER). The RER should be lower than 1.0., as all the participants were exercising lower than the ventilatory threshold intensity. According to Silva et al. [14], these variables were used to calculate the oxygen cost and the energy cost of running. 

### 2.6. Cardiorespiratory Maximal Treadmill Test

After a 30-min recovery period from the running economy test, which was enough to return all volunteers’ heart rates to rest levels, they were subjected to the cardiorespiratory maximal treadmill test. The same computer-based metabolic analyzer (Quark, Comedy, Italy) was used to measure V̇O_2_max, VT, and respiratory compensation point (RCP). The maximal aerobic speed (MAS) was also measured. V̇O_2_max was defined as a stable increase in oxygen uptake (less than 2.1 mL/kg/min) even after increasing exercise intensity [15]. VT was calculated using the following criteria: An increase in the ventilatory equivalent for oxygen without an increase in the equivalent for carbon dioxide and an increase in end-tidal pressure of oxygen. The RCP was determined by increasing the CO_2_ equivalent ventilatory and decreasing the end-tidal pressure of CO_2_ [16]. Two independent investigators determined VT and RCP, and a third researcher was consulted in the event of disagreement. The MAS was defined as the lowest exercise intensity that produced V̇O_2_max [10].

### 2.7. Total Race and Split Time Results

Total race and split time results were provided by the organizers of the 30th Santos International Olympic Triathlon, which were accessed via the official website of the event (https://www.internacionaldesantos.com.br/, accessed on 5 March 2022).

### 2.8. Statistical Analysis

The data were presented in the form of mean and standard deviations. According to the Kolmogorov–Smirnov and Levene’s tests, all variables had a normal distribution and homogeneous variability. A Student’s *t*-test for independent samples was used to compare the variables of male and female athletes. To determine the magnitude of the differences, between group effect sizes were computed for each outcome. Using Cohen’s effect sizes, the magnitude of any change was judged according to the following criteria: *d* < 0.2 was considered as ignored; 0.2 ≤ *d* < 0.5 was considered as a “small” effect size; 0.5 ≤ *d* < 0.8 represented a “moderate” effect size; 0.8 ≤ *d* < 1.3 a “large” effect size; and *d* ≥ 1.3 a “very large” effect [17]. The Pearson linear correlation coefficient and dispersion diagrams were used to validate the level of association between each split time and total race time with other measured variables. To compare the triathlon race time to the previous triathlon experience, a one-way ANOVA was used. Thereafter, we considered the significant correlations for the stepwise adjustment of the multiple linear regression model. The formula of the regression model is x = α + β·y + E, where x is the dependent variable, y is the independent variable, α is the intercept, β is the slope, and E is the residual. For each regression equation, the coefficient of determination (r^2^), a number that measures how well a statistical model predicts an outcome, was presented. For all regression models presented, Durbin–Watson Test (to detect autocorrelation), variance inflation factor (VIF) and tolerance (to detect multicollinearity), the normality of the distribution of residuals, and Q-Q plot (to detect homoscedasticity) were presented. The G*Power version 3.1.9.2 (Franz, Universität Kiel, Germany) was used to determine the sample size and analyze the test power level. A sample size calculation for regression analysis for overall race time with two predictors, using previous published data from Puccinelli et al. [10] (r^2^ = 0.607), showed that 20 athletes were needed to detect a relevant difference with 80% power and a significance level of 5%. The powers of the analyses were also calculated. The analyses were carried out using the IBM SPSS Statistics (version 22, USA) software, with the level of significance set at *p* < 0.05.

## 3. Results

In terms of physiological variables, men had significantly higher values for V̇O_2_max (L/min) (*p* < 0.001, *d* = 2.74), V̇O_2_max (mL/kg/min) (*p* = 0.020, *d* = 0.769), MAS (*p* < 0.001, *d* = 1.30), VT speed (*p* = 0.011, *d* = 0.843), and RCP speed (*p* < 0.001, *d* = 1.78). The results showed no significant difference in the percentage of V̇O_2_max at VT (*p* = 0.129, *d* = 0.490) or RCP (*p* = 0.558, *d* = 0.173) between men and women. Running economy, as measured by oxygen cost or energy cost, did not differ significantly between sexes (*p* = 0.540, *d* = 0.196 and *p* = 0.600, *d* = 0.167, respectively) (Table 2).

Men were significantly faster in the swimming split (*p* = 0.041, *d* = 0.670) and cycling split (*p* = 0.013, *d* = 0.826) of the Olympic-distance triathlon race, but there was no significant difference in the running split (*p* = 0.608, *d* = 0.163) or total race time (*p* = 0.078, *d* = 0.573) (Table 2). There was no significant difference in the years of triathlon experience (*p* = 0.807) between women [3(1–3)] and men [3(1–3)]. The level of association between each split and total race performance with body composition or physiologic variables was presented in Table 3.

When female (*p* = 0.113) and male (*p* = 0.217) athletes with less than 1 year, 1–3 years, and more than 3 years of triathlon training experience were compared, no significant difference in the triathlon race time was found. There was also no significant difference in swimming, cycling, and running split times between those with less than 1 year, 1–3 years, and more than 3 years of triathlon training experience in the female group (*p* = 0.053, *p* = 0.397, *p* = 0.137, respectively) and male group (*p* = 0.077, *p* = 0.248, *p* = 0.380, respectively).

Multiple linear regression adjusted models were fitted to determine which measured body composition and physiological variables can better predict the results by the stepwise method in each split time and the overall race time for women and men. The statistical models resulting from the analyses for the female sample are presented in Table 4, while those for the male sample are presented in Table 5.

For the women, in swimming, the variables that better adjusted to the model were absolute V̇O_2_max (β = −877, t = −4.50, *p <* 0.001) and experience in triathlon competition (β = −378, t = −245, *p* = 0.026), and both explain 63.4% of the split time performance in swimming. In cycling, the best model used only one variable: V̇O_2_max relative to body mass (β = −41.8, t = −5.31, *p* < 0.001), and it can explain 61% of the cycling split time. Similarly, the best model for running used only one variable: The speed at RCP (β = −325, t = −6.62, *p* < 0.001), and it can explain 70.9% of the running split time. Finally, for total race time, the best model includes V̇O_2_max (β = −131, t = −6.61, *p* < 0.001), lean mass (β = −61.4, t = −2.66, *p* = 0.018), and triathlon experience (β = −886.1, t = −3.01, *p* = 0.009), and they can explain 82.5% of the overall race time (Table 4).

For the men, in swimming, the best variables were speed at RCP (β = −89.6, t = −2.31, *p* = 0.034) and percentage of android fat (β = 22.0, t = 2.50, *p* = 0.023), and both can explain 49.4% of the swimming performance. In cycling, the best equation used only one variable: Percentage of android fat (β = 36.1, t = 3.16, *p* = 0.005), and it can explain 35.7% of the cycling split time. Similarly, in running, the best model used only one variable: Speed at RCP (β = −325, t = −6.88, *p* < 0.001), and it can predict 72.4% of the running performance. For the total race time, the variables that best adjusted to the model were MAS (β = −294.1, t = −2.89, *p* = 0.010) and percentage of body fat (β = 53.6, t = 2.20, *p* = 0.042), and both can predict 57.8% of the performance (Table 5).

## 4. Discussion

The main findings of this study were as follows: (I) The variables that better adjusted to the regression models for triathlon performance were different for male and female athletes; (II) for women, V̇O_2_max was part of the prediction equations for performance in swimming, cycling, and overall race time; (III) for women, the triathlon experience time was part of the prediction equations for performance in swimming and overall race time; (IV) for women, the lean mass was part of the prediction equations for the overall race time; (V) for men, body composition (android fat mass percentage or body fat percentage) was part of the prediction equations for performance in swimming, cycling, and overall race time; and (VI) the model for predicting performance in running split, which is the speed at RCP, was the only one that was found to be very similar between sexes. The current study’s findings confirmed this initial hypothesis, as the regression models for performance in swimming, cycling, and overall race time differed between sexes.

### 4.1. Physiological, Body Composition, and Performance Sex Differences

In terms of body composition, the male sample had significantly more lean mass than the female sample. However, there was no difference in fat mass (kg) between sexes. The % fat mass was also not significantly different (*p* = 0.066) between sexes; however, the effect size of the difference was 0.6 (medium effect), and the power of this analysis was very low (power = 0.453); therefore, caution should be exercised when claiming that there is no difference in % fat mass between sexes. Despite the similarity of the total fat mass (kg), the distribution of body fat differed significantly between sexes. Female athletes had a higher gynoid percentage, while male athletes had a higher android percentage. These data are consistent with the previous study [18] for non-athletes and athletes [19]. Moreover, these findings are especially important given that the regional fat distribution in gynoid or android regions is associated with performance [19] and lipid profile [18].

In terms of physiological variables, male athletes had higher V̇O_2_max and MAS than female athletes. However, the percentage of V̇O_2_max at VT and RCP did not differ between sexes. Although the literature agrees that men have higher maximum oxygen consumption values [8], data on VTs are contradictory. Puccinelli et al. [10] discovered that female athletes had a higher percentage of V̇O_2_max at VT and RCP than male athletes. These contradictory findings could be attributed to the varying levels of training of the athletes. Moreover, Puccinelli et al. [10] reported that the V̇O_2_max values for male athletes were 59.9 ± 6.3 mL/kg/min and our male sample had a lower value of 54.6 ± 5.0 mL/kg/min, whereas female V̇O_2_max values were comparable between the two studies (49.5 ± 7.8 and 49.7 ± 7.6 mL/kg/min, respectively).

The running economy did not differ between sexes. This finding is consistent with previous literature data [4]. Men outperformed women in cycling and swimming, with the difference being more evident in cycling. In the running split and the overall race time, there was no significant difference between sexes; however, due to the fact that the power of this statistical analysis was low (power = 0.079; power = 0.423, respectively), caution is advised in interpreting this lack of significant difference between sexes.

### 4.2. Predictors of Triathlon Overall and Split Race Times for the Female Sample

For the female sample, absolute (L/min) and relative (mL/kg/min) V̇O_2_max values are strongly related to all the split and overall race times. This is an expected result since the higher the individual’s oxygen consumption capacity, the greater the exercise intensity they can sustain [9,10]. The V̇O_2_max by the regression models for swimming split time, cycling split time, and overall split time, demonstrate the importance of this variable in female performance.

Triathlon experience appears to be an important variable in predicting performance for female athletes, as evidenced by the prediction equations for swimming split time and overall race time. Moreover, previous research has shown that triathlon experience is an important predictor of performance in splits, particularly in swimming and total race time [9,20,21,22,23]. Swimming in the sea, bay, lake, or river (current, temperature, navigation, buoyancy, etc.) can have very different environmental conditions than swimming in a pool, where most athletes train, and these differences can make triathlon training experience especially important for swimming performance. Previous studies have already demonstrated the positive impact of previous experience on performance in the triathlon’s total race time [20,21]. In the same direction, sports practice during childhood, even non-specific physical activity, has previously been shown to be highly correlated to better performances in swimming and total race times in the Olympic triathlon [24].

Apart from swimming, regarding body composition, body fat mass (kg or %) showed a significant correlation with cycling and running split time and total race time for women. These findings are consistent with previous literature data [5,9,25]. The lack of association between body fat and swimming split time could be attributed to fatter individuals having better body buoyancy due to lower body density, which contributes to passive floating and gliding [25,26]. Even though there is a significant association between body fat and running or cycling performance, it is not included in the regression models. The absence of body fat in the equation can be explained by the fact that this variable is associated with V̇O_2_max, and since V̇O_2_max composes the equation, body fat could also not compose the equation to avoid the multicollinearity effect.

The cycling performance was strongly associated with lean mass. The important relationship that exists between muscle mass and cycling performance can be explained by a greater ability to apply force on the pedals, which will generate greater power and displacement speed in cycling [27]. In contrast, the present study found no significant association between muscle mass and swimming split performance. Despite the fact that muscle mass contributes to propulsive force during swimming, it also contributes to an increase in body density and the tendency for body sinking, which increases swimming drag [28,29]. Therefore, when considering swimming speed as the function of drag and propulsion forces, there is no agreement in the previous literature on whether muscle mass has a positive or negative effect on swimming performance [24,26].

In terms of overall race time, muscle mass appears to be very important for female athletes, as this variable composes the regression model used to predict the overall race time. The speed at RCP, MAS, and V̇O_2_max (mL/kg/min) represented the measured variables that showed the strongest association with running performance, which is consistent with previous literature data [22,30,31]. Regarding the regression model for running split time, the speed at RCP has been used to build the model.

### 4.3. Predictors of Overall Triathlon and Each Split Race Time for the Male Sample

Regarding the physiological variables, the V̇O_2_max (mL/kg/min) presented a significant correlation with running split time and overall split time, similar to the MAS. At the same time, the speed and V̇O_2_ measured at the VT and RCP had a significant correlation with swimming and running split times, as well as overall race time. The significance of these variables can be seen in the fact that the speed at RCP composes the regression models used to predict the swimming and running time, whereas MAS composes the model used to predict the overall race time. The inclusion of the MAS rather than the V̇O_2_max in the prediction model of the overall race time can be justified since the MAS is known to be dependent on the V̇O_2_max and running economy [10]. The lack of association of V̇O_2_max and VT measurements with cycling performance may be due to some factors, such as the lack of specificity of the assessment method. The test used to determine V̇O_2_max and VTs was developed on a treadmill, and it is possible that an assessment performed on a bicycle will be more effective in identifying the variables associated with cycling split performance. Another complicating factor to consider is the tactical dimension in cycling split, such as drafting, pacing, and contextual factors on race dynamics [32]. Moreover, the results showed no significant association between running economy and performance. It is already expected, since running economy is regarded as a particularly important variable for distinguishing performance only among athletes with similar values for V̇O_2_max [33], which is not the case in the present study.

In terms of body composition variables, lean mass (kg) was not related to any split or overall race time. In contrast, fat mass (%) was associated with swimming, running, and overall race time. In addition, the fat mass percentage was used to create regression models to predict overall race time. Even though the total fat mass (%) was not associated with cycling split time, the distribution of fat mass affects performance in cycling and swimming split times. The higher the percentage of fat mass in the android, the worse the performance. With increased abdominal volume, it can be more difficult to assume an aerodynamic posture on a time trial bike, which can impair cycling performance. In a time trial bike position, the increase in abdominal volume can also make breathing mechanics difficult, where small shifts in cyclist positions may have a meaningful impact on performance [34]. A higher android fat percentage can also impair optimal body posture while swimming. McLean et al. [26] demonstrated that a higher android fat mass contributes to decreased buoyancy, which increases drag forces and reduces swimming speed [25]. Therefore, fat mass distribution is a very important variable associated with performance, and it was used to construct the regression model for swimming and cycling split times.

The regression model used in this study to predict swimming split time was composed of RCP speed and android fat mass (%). The android fat mass was the unique variable that composed the regression model for cycling split time. It is possible that evaluating the physiological variables in a more specific ergometer (bike) will yield regression models with better predictive values. In this direction, for the running split time, only the RCP speed is used to build the regression model. Finally, MAS and total body fat (%) composed the regression model for overall race time. As a result, it is demonstrated that only a treadmill assessment of MAS and a body composition assessment can predict more than 50% of the overall race time in an Olympic-distance triathlon.

### 4.4. Limitations and Strengh of the Study

The lack of physiological measurements in swimming or cycling activities is one of the study’s limitations. It is possible that with these additional evaluations (cardiorespiratory maximal tests performed at cycle ergometer or at swimming pool), better equations for predicting triathlon performance can be presented. The authors propose that future studies should be designed with this goal in mind. The presentation of triathlon performance prediction equations for female athletes is an important strength of the study, as women are understudied, and the factors associated with performance differ between sexes. In addition, the sample size can be considered as very adequate since it reached a high level of power. As the power of a hypothesis test is 1 minus the probability of a type II error, the probability of making a type II error was exceedingly small. Finally, another strength of the study was using reliable and valid instruments, such as DEXA and breath, via the breath gas exchange analyzer.

## 5. Conclusions

For women’s endurance performance, there are strong correlations between physiological variables usually measured in a laboratory. Moreover, it was possible to develop significantly predictive performance equations for triathlon total race time and splits. The physiological measures evaluated in the incremental treadmill test and body composition variables can predict more than 50% of the performance in the total time of the Olympic triathlon event. In addition, regression models for predicting female performance can predict a higher percentage of performance than models for predicting male performance. Finally, and perhaps most importantly, the variables capable of predicting male performance are not the same as those capable of predicting female performance, justifying the need to evaluate and study each gender separately.

## Figures and Tables

**Table 1 healthcare-11-00622-t001:** Anthropometric data for men and women.

	Women (n = 20)	Men (n = 20)	*p*-Value	Cohen’s *d*	Power
Age (years)	42.7 ± 7.3 (3.9–45.9)	43.7 ± 9.3 (38.2–46.4)	0.880	0.05	0.052
Body mass (kg)	58.8 ± 6.7 (55.9–61.8)	74.8 ± 6.9 (71.8–77.9)	<0.001	2.35	0.999
Height (cm)	165.0 ± 5.7 (163.0–168.0)	175.0 ± 8.2 (171.0–178.0)	<0.001	1.35	0.986
Fat mass (kg)	13.3 ± 7.2 (9.9–16.9)	12.8 ± 5.0 (10.6–15.0)	0.826	0.07	0.055
Lean mass (kg)	42.2 ± 6.5 (45.1–39.4)	58.3 ± 5.8 (55.8–60.8)	<0.001	2.60	1.000
% Body fat	23.3 ± 11.3 (18.3–28.2)	17.8 ± 6.3 (15.0–20.6)	0.066	0.60	0.453
% Gynoid fat	53.2 ± 6.0 (50.6–55.9)	39.1 ± 5.1 (36.8–41.3)	<0.001	2.53	1.000
% Android fat	42.9 ± 6.6 (40.0–45.8)	57.1 ± 5.4 (54.7–59.5)	<0.001	2.36	0.999

Mean ± standard deviation. Confidence interval: 95%.

**Table 2 healthcare-11-00622-t002:** Measured variables in the cardiorespiratory maximal test and performance for each sex.

	Women (n = 20)	Men (n = 20)	*p*-Value	Cohen’s *d*	Power
**Cardiorespiratory maximal test**					
V̇O_2_max (L/min)	2.90 ± 0.39 (2.72–3.07)	4.08 ± 0.47 (3.88–4.28)	<0.001	2.74	1.00
V̇O_2_max (mL/kg/min)	49.7 ± 7.6 (46.3–53.0)	54.6 ± 5.0 (52.4–56.8)	0.020	0.769	0.659
MAS (km/h)	14.9 ± 1.8 (14.1–15.7)	17.1 ± 1.6 (16.4–17.8)	<0.001	1.30	0.979
V̇O_2_ at VT (mL/kg/min)	38.0 ± 6.3 (35.3–40.0)	40.4 ± 4.0 (38.6–42.2)	0.164	0.499	0.336
% V̇O_2_max at VT	76.2 ± 5.3 (73.9–78.6)	73.8 ± 4.7 (71.7–75.9)	0.129	0.490	0.326
Speed at VT (km/h)	10.7 ± 1.6 (10.0–11.4)	11.8 ± 1.1 (11.4–12.3)	0.011	0.843	0.738
V̇O_2_ at RCP (mL/kg/min)	45.1 ± 6.9 (42.1–48.1)	49.1 ± 4.7 (47.4–51.1)	0.025	0.736	0.621
% V̇O_2_max at RCP	90.5 ± 3.9 (88.7–92.2)	89.8 ± 3.7 (88.2–91.4)	0.588	0.173	0.083
Speed at RCP (km/h)	12.8 ± 1.6 (12.0–13.5)	14.4 ± 1,2 (13.9–15.0)	<0.001	1.78	0.999
Oxygen cost (mL/kg/km)	226.0 ± 20.3 (217.0–235.0)	222.0 ± 18.1 (214.0–230.0)	0.540	0.196	0.092
Energy cost (Kcal/kg/km)	1.11 ± 0.09 (1.07–1.15)	1.09 ± 0.09 (1.05–1.13)	0.600	0.167	0.080
**Race performance**					
Swimming (seconds)	2072 ± 518 (1844–2219)	1796 ± 265 (1680–1912)	0.041	0.670	0.670
Cycling (seconds)	4274 ± 405 (4097–4452)	3969 ± 329 (3825–4114)	0.013	0.826	0.720
Running (seconds)	3143 ± 546 (2904–3382)	3060 ± 371 (2853–3266)	0.608	0.163	0.079
Total race time (seconds)	9489 ± 1357 (8894–10083)	8825 ± 920 (8421–9228)	0.078	0.573	0.423

Mean ± standard deviation. Confidence interval: 95%. MAS: Maximal aerobic speed; V̇O_2_ at VT: V̇O_2_ at ventilatory threshold; % V̇O_2_max at VT: % V̇O_2_max at ventilatory threshold; V̇O_2_ at RCP: V̇O_2_ at respiratory compensation point; % V̇O_2_max at RCP: % V̇O_2_max at respiratory compensation point.

**Table 3 healthcare-11-00622-t003:** Pearson’s correlation coefficient between performance in swimming, cycling, running, total race time, and measured variables for both sexes.

	Swimming (n = 20)	Cycling (n = 20)	Running (n = 20)	Total Time (n = 20)
Fat mass (kg)	W: r = 0.278 M: r = 0.475 *	W: r = 0.604 * M: r = 0.245	W: r = 0.612 * M: r = 0.657 *	W: r = 0.533 M: r = 0.560 *
Lean mass (Kg)	W: r= −0.345 M: r= −0.257	W: r= −0.530 * M: r= −0.210	W: r= −0.413 M: r= −0.382	W: r= −0.456 * M: r= −0.344
% Body fat	W: r = 0.353 M: r = 0.495 *	W: r = 0.653 * M: r = 0.300	W: r = 0.648 * M: r = 0.702 *	W: r = 0.590 * M: r = 0.609 *
% Gynoid fat	W: r = 0.188 M: r= −0.541 *	W: r= −0.037 M: r= −0.597 *	W: r= −0.035 M: r= −0.258	W: r = 0.046 M: r= −0.501 *
% Android fat	W: r= −0.138 M: r = 0.580 *	W: r = 0.108 M: r = 0.598 *	W: r = 0.110 M: r = 0.353	W: r = 0.024 M: r = 0.561*
V̇O_2_max (L/min)	W: r= −0.713 * M: r= −0.169	W: r= −0.776 * M: r= −0.154	W: r= −0.711 * M: r= −0.361	W: r= −0.790 * M: r= −0.288
V̇O_2_max (mL/kg/min)	W: r= −0.634 * M: r= −0.375	W: r= −0.781 * M: r= −0.237	W: r= −0.828 * M: r= −0.678 *	W: r= −0.808 * M: r= −0.539 *
MAS (km/h)	W: r= −0.633 * M: r= −0.442	W: r= −0.690 * M: r= −0.406	W: r= −0.845 * M: r= −0.790 *	W: r= −0.788 * M: r= −0.676 *
V̇O_2_ at VT (mL/kg/min)	W: r= −0.659 * M: r= −0.475 *	W: r= −0.794 * M: r= −0.343	W: r= −0.802 * M: r= −0.614 *	W: r= −0.811 * M: r= −0.573 *
% V̇O_2_max at VT (%)	W: r= −0.166 M: r= −0.278	W: r= −0.141 M: r= −0.235	W: r= −0.089 M: r= −0.036	W: r= −0.141 M: r= −0.182
Speed at VT (km/h)	W: r= −0.490 * M: r= −0.591 *	W: r= −0.701 * M: r= −0.385	W: r= −0.787 * M: r= −0.803 *	W: r= −0.713 * M: r= −0.719 *
V̇O_2_max at RCP (mL/kg/min)	W: r= −0.581 * M: r= −0.548 *	W: r= −0.757 * M: r= −0.298	W: r= −0.793 * M: r= −0.695 *	W: r= −0.581 * M: r= −0.620 *
% V̇O_2_max at RCP	W: r = 0.208 M: r= −0.386	W: r = 0.107 M: r= −0.240	W: r = 0.149 M: r= −0.050	W: r = 0.171 M: r = 0.224
Speed at RCP (km/h)	W: r= −0.576 * M: r= −0.556 *	W: r= −0.731 * M: r= −0.416	W: r= −0.842 * M: r= −0.851 *	W: r= −0.777 * M: r= −0.744 *
Oxygen cost (mL/kg/min)	W: r= −0.065 M: r = 0.034	W: r= −0.510 * M: r= −0.60	W: r = 0.292 M: r = 0.141	W: r= −0.294 M: r = 0.061
Energy cost (kcal/kg/km)	W: r = 0.014 M: r = 0.070	W: r= −0.453 * M: r= −0.040	W: r= −0.207 M: r = 0.200	W: r= −0.213 M: r = 0.109

W: Women; M: Men; r: Pearson’s correlation coefficient; *p*-value between parenthesis; * *p* < 0.05; mean ± standard deviation. Confidence interval: 95%. MAS: Maximal aerobic speed; V̇O_2_ at VT: V̇O_2_ at ventilatory threshold; % V̇O_2_max at VT: % V̇O_2_max at ventilatory threshold; V̇O_2_ at RCP: V̇O_2_ at respiratory compensation point; % V̇O_2_max at RCP: % V̇O_2_max at respiratory compensation point.

**Table 4 healthcare-11-00622-t004:** Multiple linear regression models for estimating performance in swimming, cycling, running, and total race time for female athletes.

Modality	r^2^	Z	Df	*p*-Value	SEE	Tolerance	VIF	Durbin– Watson	Power
	Time 1500 m (s) = 4817 − 877 (absol. V̇O_2_max) − 378 (T.E.)								
Swimming	0.634	13.8	2.16	<0.001	304	>0.997	1	2.17	0.99
	Time 40 Km (s) = 6347 − 41.8 (relat. V̇O_2_max)								
Cycling	0.610	28.2	1.18	<0.001	246	0.961	1	1.59	0.99
	Time 10 Km (s) = 6766 − 284 (speed at RCP)								
Running	0.709	43.8	1.18	<0.001	287	1	1	2.35	0.99
	Total time (s) = 19077.4 − 131 (relat. V̇O_2_max) − 61.4 (lean mass) − 886.1 (T.E)								
Total Time	0.825	23.6	3.15	<0.001	561	1	<1.08	2.15	1.00

Absol. V̇O_2_max: Absolute V̇O_2_max (L/m); T.E.: Triathlon experience > 3 years; Relat. V̇O_2_max: V̇O_2_max relative to body mass (mL/kg/min); speed at RCP: Speed at respiratory compensation point.

**Table 5 healthcare-11-00622-t005:** Multiple linear regression models for estimating performance in swimming, cycling, running, and total race time for men.

Modality	R^2^	Z	Df	*p*-Value	SEE	Tolerance	VIF	Durbin– Watson	Power
	Time 1500 m (s) = 1836.8 − 89.6 (speed at RCP) + 22 (% Android fat)								
Swimming	0.494	8.31	2.17	0.003	183	0.907	1.1	1.82	0.95
	Time 40 Km (s) = 1907.2 + 36.1 (% Android fat)								
Cycling	0.357	10	2.18	0.005	257	1	1	2.17	0.88
	Time 10 km (s) = 7750 − 325 (speed at RCP)								
Running	0.724	47.3	1.18	<0.001	241	1	1	1.71	0.99
	Total time (s) = 12850 − 294.1 (MAS) + 56.3 (% Body Fat)								
Total Time	0.578	11.7	2.17	<0.001	582	0.808	<1.24	1.60	0.99

Speed at VT-2: Speed at ventilatory threshold-2 (km/h); MAS: Maximal aerobic speed (km/h).

## Data Availability

Data supporting the reported results can be requested from the corresponding author.

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
