# Peer review of "Physiological Features of Olympic-Distance Amateur Triathletes, as Well as Their Associations with Performance in Women and Men: A Cross–Sectional Study"

_healthcare, 2023, doi:10.3390/healthcare11040622_

Round 1

Reviewer 1 Report

Firstly, I would like to congratulate the authors for their sensitivity in apporachingf their work with a gender perspective. With studies like this, an important contributios is made to solving the gender gap in sport.

Secondly, I would like to highlight the quality of the study, as well as its relevance to the scientific context.

However, and with the intention of contribuiting to the improvemente of the work, I will make a series of evaluations. I suggest they be reviewed:

(i) Table 1 shows the descriptive data of the sample. I don´t understand why differences are calculates between the variables. What is the intention of this? Similarly, I don´t share the first 4 lines of results. They are part of the description of the sample..., but not of the results. same with discussion

(ii) I recommend, in the statistical analysis, to include the formula of the regression model.

(iii) In the discussion, in the last paragraphs of sections4.2 and 4.3, data are presented that I consider to be part of the results. I recommend a review of the content location.

Author Response

Reviewer #1

Firstly, I would like to congratulate the authors for their sensitivity in approaching their work with a gender perspective. With studies like this, an important contributions is made to solving the gender gap in sport.

Secondly, I would like to highlight the quality of the study, as well as its relevance to the scientific context.

Answer: Thank you for your positive comment.

However, and with the intention of contributing to the improvement of the work, I will make a series of evaluations. I suggest they be reviewed:

Answer: Thank you for your suggestions.

(i) Table 1 shows the descriptive data of the sample. I don´t understand why differences are calculates between the variables. What is the intention of this? Similarly, I don´t share the first 4 lines of results. They are part of the description of the sample..., but not of the results. same with discussion

Answer: Thank you about your constructive comments. Firstly, our intention to compare sexes in Table 1 is to demonstrate that the age of the male and female athletes was not different. We think it is an important data, once we are comparing some physiological variables in Table 2, and if the ages were different, it would be a bias of the study. In addition, although the body composition sex differences have already been previous presented by the literature, we consider it is an important a positive control of the study and shows that the recruited participants had expected characteristics due to the results traditionally presented in the literature. For the same reason, we decided to maintain the discussion about the sex differences. To clarify this aim, we decided to include this aim in the introduction section.

We agree with the expert reviewer, and the first 4 line of results have been excluded and the reference to Table 1 is presented in the participant’s section. Please let us know if these changes do not resolve your doubts.

(ii) I recommend, in the statistical analysis, to include the formula of the regression model.

Answer: The formula of the regression model is x = a + b.y + E, where x is the dependent variable, y is the independent variable, a is the intercept, b is the slope, and E is the residual. This information has been included in the manuscript in order to clarify.

(iii) In the discussion, in the last paragraphs of sections 4.2 and 4.3, data are presented that I consider to be part of the results. I recommend a review of the content location.

Answer: We agree with the expert reviewer and the content has been transferred to the results section in order to meet with your expectation. Please let us know if these changes do not resolve your doubts.

Reviewer 2 Report

well written

please check spell and grammar throughout the manuscript

mention ethical committee approval number and date

remove very old references and add recent references

Author Response

Reviewer #2

well written

Answer: Thank you for your positive comment.

please check spell and grammar throughout the manuscript

Answer: was checked again

mention ethical committee approval number and date

Answer: Thank you for drawing our attention to this point. The approval number is 5.059.538, October 25, 2021. This information has been included in Line 94.

remove very old references and add recent references

Answer: Thank you your constructive comment. The old references have been replaced to new references everywhere it was possible.

Reviewer 3 Report

Dear authors,

Please review the document.

Thanks.

Author Response

Reviewer #3

Dear authors, It is a study that presents a relative novelty in the field of sports training research. However, the work they have done to assess performance in triathlon and performance prediction models for cycling, swimming and running is interesting. Therefore, an adequate methodology has been carried out to meet the objectives.

Answer: Thank you for your suggestions and insightful comments.

Some considerations

  1. This idea has already been expressed in the regression results (line 35 36, Summary section).

Answer: Thank you for your commentary. We reviewed the abstract but chose to respectfully disagree with the reviewer. Despite the results showing that the variables that compose the regression models are different, we understand that drawing attention to this fact in the conclusion is very important, as this issue was a central problem of the study.

  1. Although he realizes the knowledge gap on the subject, it is necessary to justify a little more why it is necessary to carry out prediction equations for women in triathlon. Who will be its direct beneficiaries, who will give practical utility to the results of this research, what is the transfer to the research field? (read 61, end of paragraph).

Answer: Thank you about your constructive commentary. The knowledge of the main predictive variables of performance for each sex separately, can help coaches to direct the training sessions to obtain adaptive responses of the most important predictive variables of performance and optimize the improvement of sports performance. This justification has been included in the manuscript in order to clarify. Please let us know if these changes do not resolve your doubts.

  1. This hypothesis can be confirmed, that is, aerobic performance and body composition variables have had different degrees of correlation, therefore, in other disciplines or sports, different equations have been developed for each gender (line 66 - 67).

Answer: There are theoretical reasons to hypothesize that the prediction equations for men's and women's regards to triathlon performance are different. This issue was the primary aim of our study. We agree with reviewer that in other sports different equations should be developed to clarify specific differences between sexes.

  1. The age range is very wide. What do the authors think? (line 74).

Answer: Despite the range of ages was wide, it reflects the participation characteristics of amateur athletes. We thought that this could be a bias if the range was different for each sex, but the range was similar for both sexes (male athletes 26 to 60 years old, and female athletes 33 to 55 years old) and there was no difference between mean ages.

  1. I do not consider them exclusion criteria, since they are eventual conditions that could occur during the competition. If this occurs, they could be considered as having incomplete data. I suggest considering only the pregnancy exclusion criteria or competing in an alternate modality (line 77 78).

Answer: We understand the reviewer's questioning, but we respectfully disagree. The initial aim of the study was to verify the association level between the variables measured in the laboratory with the performance in the race, therefore we excluded from the study volunteers that did not complete the data collection, and therefore could not be included in the data analysis. Since this criterion was established before the beginning of the study, we chose to maintain it.

  1. It must be explicit that the participants signed the informed consent (line 91).

Answer: Thank you for drawing your attention to this point. The sentence has been rewritten and the information has been included.

  1. It must be made clear in how many sessions all the measurements were made and the hours or days of rest between them (line 94 96).

Answer: Thank you about your commentary. All the tests were made at the same day, during the morning period. During the visit, volunteers were measured for height, body mass, and body composition. After that they were submitted to a running economy evaluation. After 30 min rest, participants were subjected to a cardiorespiratory maximal test. The sequence of tests has been explained in the manuscript.

  1. Change to black text color (line 98 100).

Answer: Thank you for drawing your attention to this point. It was changed.

  1. Indicate make, madel, country (line 109 “stadiometer”)

Answer: Thank you for drawing your attention to this point. The stadiometer characteristics has been included.

  1. Clarify if it is a gas exchange analyzer (ergospirometer) (line 117 118).

Answer: Thank you for drawing your attention to this point. Yes, it is a breath-by-breath gas exchange analyzer. This information has been included in the text in order to clarify.

  1. Clarify whether they are referring to the respiratory exchange rate (RER) and what the criterion value was (line 121 -122 “respiratory coeficients”)

Answer: Thank you about your comment. The sentence has been rewritten as requested by you.

  1. It's a letter "O", not a number zero "0” (line 128 and all text: “VO2max”

Answer: Thank you about your comment. VO2max has been replaced in all text.

  1. Use black color (line 146 147).

Answer: Thank you for drawing your attention to this point. It was changed.

  1. Do the authors consider it important to check the degrees of agreement of the regression models using the Bland-Altman diagram? Justify your answer (line 254).

Answer: A Bland-Altman diagram is a very useful instrument to visualize the differences in measurements between two different instruments or two different measurement techniques. Considering that we did not use two different instruments or techniques for any measurement, we understand that the Bland Altman diagram would not be useful for this study. We emphasize that to guarantee the quality of the regression models presented, Durbin–Watson Test (to detect autocorrelation), variance inflation factor (VIF) and tolerance (to detect multicollinearity), the normality of the distribution of residuals, and Q-Q plot (to detect homoscedasticity) were presented. Please let us know if these explanations do not resolve your doubts.

  1. In the swimming model, it could be interpreted that when the VO2max. absolute increases by 1 L/min, the time to complete the 1500 meters decreased by 887 seconds. What do the authors think of this interpretation? Could the other models for women and men be interpreted in the same way? (Table IV, line 208).

Answer: Yes, when the VO2max. absolute increases by 1 L/min, the time to complete the 1500 meters decreased by 887 seconds. This effect of VO2max on swimming time may seem very large, but it is important to remember that the VO2max that composes this model is the absolute value (L/min) and that changes of 1L are very large. For example, a men (70kg), VO2max = 2L/min, presents VO2max adjusted to body weight = 42ml/kg/min, but if the same men (70kg) presents VO2max = 3L/min, he presents VO2max adjusted to body weight = 57 ml/kg/min, which is a huge difference and explains the big impact on swimming performance.

  1. Put in black color (“power”, table IV and table V)

Answer: Thank you for drawing your attention to this point. It was changed.

  1. What do the authors think that neither model included the VO2max. relative or absolute as an independent variable? (Table V).

Answer: The maximal oxygen uptake (VO2max) is a variable that is strongly associated with performance, which can be seen in Table 3.  The importance of this variable occurs because the greater the athlete's capacity to consume oxygen, the faster he will be able to run (cycle or swim) aerobically. However, what matters for performance is speed sustained during the race. One athlete may eventually have a high VO2max value, but not be able to run fast if he has poor running economy. For this reason, we believe that MAS and the speed at RCP composed the models and not the VO2max. However, it is worth remembering that to run fast, an athlete needs to present high VO2max value and the two variables were not included in the regression model to avoid the muticolinearity effect.

  1. “Body fat” , Table V, total time equation.

Answer: Thank you for drawing your attention to this point. It is corrected.

  1. This is what I was referring to in the comment in the introduction section on line 66 (line 233).

Answer: Yes, we think it is important to conclude that the variables with impact the male performance were not the same as the variables that mostly impact the female performance. Therefore, the training section aiming to improve the performance should not be equal for both sexes.

  1. Include point after parenthesis (line 280).

Answer: Thank you for drawing your attention to this point. It is corrected.

  1. On occasions, a bivariate correlation is not necessarily statistically significant within a multivariate model (line 311 322).

Answer: The sentence has been excluded and the data are transferred to the results section, as requested by the reviewer #2

  1. which ones? (line 365 “additional evaluations”)

Answer: For example: cardiorespiratory maximal test performed at cycle ergometer or at swimming pool). The sentence has been rewritten.

  1. A strength was using reliable and valid instruments such as DEXA, ergospirometer... (line 368).

Answer: Thank you for drawing your attention to this point. It is included.

  1. Put in black color (line 370 372).

Answer: Thank you for drawing your attention to this point. It was changed.

  1. At the same time, it is a limitation of the study to develop regression models with only 10 participants for each sex (line 370).

Answer: The study was developed with 20 participants for each sex, and according to the sample size calculation performed prior to the study it is enough for α = 0.05 and β= 0.80.

Round 2

Reviewer 3 Report

Dear Authors, 

Some answers that are not clear.

11. Clarify whether they are referring to the respiratory exchange rate (RER) and what the criterion value was (line 121 – 122, “respiratory coefficients”).

Author response: The RER should be lower than 1.0., as all the participants were exercising lower than the ventilatory threshold intensity.

Reviewer Response: The RER must be greater than 1.11 for a VO2max             check.

12. It's a letter "O", not a number zero "0" (Line 128 and all text: “VO2max”).

              Must be a letter "O" throughout the text.

Thanks very much.

Author Response

Title: Physiological features of Olympic-distance amateur triathletes, as well as their associations with performance in women and men. A cross – sectional study

13-February-2023

Dr. Rahman Shiri

Editor-in-Chief

Dear Editor,

Thank you about the opportunity of resubmitting the revised version of the manuscript. We are sending a point-by-point answer to the comments. The only change made to the manuscript was to eliminate the italicized letter from VO2max. We hope it is now suitable for publication in the Healthcare.

Reviewer #3

Dear Authors, 

Some answers that are not clear.

  1. Clarify whether they are referring to the respiratory exchange rate (RER) and what the criterion value was (line 121 – 122, “respiratory coefficients”).

Author response: The RER should be lower than 1.0., as all the participants were exercising lower than the ventilatory threshold intensity.

Reviewer Response: The RER must be greater than 1.11 for a VO2max             check.

Answer: Thank you for your comment and for the opportunity to clarify. RER was used as a criterion for running economy test and not for maximal treadmill test. In this context, RER had to remain below 1.0 during the submaximal running test for data to be included into the analysis. This criterion agrees with previous published studies about running economy (Mendonca et al 2020, Faelli et al 2021).

We would also like to clarify that the criterion used to determine V̇O2max during the maximal treadmill test was a stable increase in oxygen uptake (less than 2.1 ml/kg/min) (V̇O2 plateau) even after increasing exercise intensity (Molinari et al 2020). As all volunteers reached the V̇O2 plateau at the end of the test, it was not necessary to use other criteria for V̇O2max determination, such as RER > 1.1.

Mendonca, G.V.; Matos, P.; Correia, J.M. Running economy in recreational male and female runners with similar levels of cardiovascular fitness. J. Appl. Physiol 2020, 129, 508–515.

Faelli, E.; Panascì, M.; Ferrando, V.; Bisio, A.; Filipas, L.; Ruggeri, P.; Bove, M. The Effect of Static and Dynamic Stretching during Warm-Up on Running Economy and Perception of Effort in Recreational Endurance Runners. Int J Environ Res Public Health. 2021 Aug 8;18(16):8386.

Molinari, C.A.; Edwards, J.; Billat, V. Maximal Time Spent at VO2max from Sprint to the Marathon. Int J Environ Res Public Health 2020, 17, 1–11.

  1. It's a letter "O", not a number zero "0"(Line 128 and all text: “VO2max”).

              Must be a letter "O" throughout the text.

Answer: Yes, it is letter “O” and not number zero. We've taken italics out from VO2max for clarity that it is letter “O” and not number zero.

Please let us know if this change does not resolve your doubts.